# Physiological and socioeconomic characteristics predict COVID-19 mortality and resource utilization in Brazil

Salomón Wollenstein-Betech[1,2], Amanda A. B. Silva[4], Julia L. Fleck[4], Christos G. Cassandras[1,2], Ioannis Ch. Paschalidis[1,2,3]*

**1** Division of Systems Engineering, Boston University, Boston, MA, United States of America, **2** Department of Electrical and Computer Engineering, Boston University, Boston, MA, United States of America, **3** Department of Biomedical Engineering, Boston University, Boston, MA, United States of America, **4** Department of Industrial Engineering, Pontifícia Universidade Católica do Rio de Janeiro, RJ, Brazil

* yannisp@bu.edu

## Abstract

### Background

Given the severity and scope of the current COVID-19 pandemic, it is critical to determine predictive features of COVID-19 mortality and medical resource usage to effectively inform health, risk-based physical distancing, and work accommodation policies. Non-clinical sociodemographic features are important explanatory variables of COVID-19 outcomes, revealing existing disparities in large health care systems.

### Methods and findings

We use nation-wide multicenter data of COVID-19 patients in Brazil to predict mortality and ventilator usage. The dataset contains hospitalized patients who tested positive for COVID-19 and had either recovered or were deceased between March 1 and June 30, 2020. A total of 113,214 patients with 50,387 deceased, were included. Both interpretable (sparse versions of Logistic Regression and Support Vector Machines) and state-of-the-art non-interpretable (Gradient Boosted Decision Trees and Random Forest) classification methods are employed. Death from COVID-19 was strongly associated with demographics, socioeconomic factors, and comorbidities. Variables highly predictive of mortality included geographic location of the hospital (OR = 2.2 for Northeast region, OR = 2.1 for North region); renal (OR = 2.0) and liver (OR = 1.7) chronic disease; immunosuppression (OR = 1.7); obesity (OR = 1.7); neurological (OR = 1.6), cardiovascular (OR = 1.5), and hematologic (OR = 1.2) disease; diabetes (OR = 1.4); chronic pneumopathy (OR = 1.4); immunosuppression (OR = 1.3); respiratory symptoms, ranging from respiratory discomfort (OR = 1.4) and dyspnea (OR = 1.3) to oxygen saturation less than 95% (OR = 1.7); hospitalization in a public hospital (OR = 1.2); and self-reported patient illiteracy (OR = 1.1). Validation accuracies (AUC) for predicting mortality and ventilation need reach 79% and 70%, respectively, when using only pre-admission variables. Models that use post-admission disease progression

**Data Availability Statement:** The data are publicly available from https://opendatasus.saude.gov.br/dataset/bd-srag-2020.

**Funding:** This research was partially supported by the National Science Foundation (NSF) in the form of grants awarded to ICP (IIS-1914792, DMS-1664644, CNS-1645681), the Office of Naval Research (ONR) in the form of a grant awarded to ICP (N00014-19-1-2571), the National Institutes of Health (NIH) in the form of grants awarded to ICP (R01 GM135930, UL54 TR004130), the National Council for Scientific and Technological Development (CNPq) in the form of a grant awarded to JLF (428907/2018-0), Coordination for the Improvement of Higher Education Personnel (CAPES) in the form of funding awarded to JLF (Finance code 001), and the Pontifícia Universidade Católica do Rio de Janeiro. The funders had no role in study design, data collection and analysis, decision to publish, or preparation of the manuscript.

**Competing interests:** The authors have declared that no competing interests exist.

information reach accuracies (AUC) of 86% and 87% for predicting mortality and ventilation use, respectively.

## Conclusions

The results highlight the predictive power of socioeconomic information in assessing COVID-19 mortality and medical resource allocation, and shed light on existing disparities in the Brazilian health care system during the COVID-19 pandemic.

## Introduction

We are experiencing a devastating global pandemic due to SARS-CoV-2, a highly infectious pathogen that causes COVID-19. Following the appearance of the first COVID-19 cases in the province of Hubei, China, in December 2019 [1], SARS-CoV-2 has infected most of the countries in the world, with over 26.6 million confirmed cases, and just under 876,000 deaths as of September 5, 2020 [2].

Several studies have identified comorbidities and clinical variables associated with higher risk of hospitalization and mortality due to COVID-19 [3–15]. Increasing evidence shows that patients with pre-existing conditions such as diabetes, lung and renal diseases, hypertension, and older age are especially at risk of succumbing to this viral infection. Additional reports have pointed to racial and ethnic differences in outcomes [16–18]. In New York City, death rates among black/African American COVID-19 patients (92.3 deaths per 100,000 population) and Hispanic/Latino (74.3) have been significantly higher than those of white (42.5) or Asian (34.5) patients [19]. In addition, an analysis of the largest integrated-delivery health system in the state of Louisiana suggested a longer wait to access care among black patients [17].

Although racial and ethnic disparities have emerged as a central topic in the conversation about COVID-19, most studies to date have assessed data from minority populations within the United States. Moreover, because data on socioeconomic status are seldom available in electronic medical record systems, the connection between socioeconomic/racial/ethnic disparities and health access inequality has yet to be investigated. It is clear, therefore, that further research on the underlying causes of COVID-19 disparities and their complex social and structural determinants is needed in order for the international scientific, public health, and clinical communities to implement interventions that alleviate excess mortality and economic disruption related to COVID-19. Because targeted public health and resource allocation policies are more effective than standard approaches [20], the design of such interventions should leverage patient subgroup-specific information, such as race/ethnicity and socioeconomic status, and be adapted to local contexts and community characteristics.

In particular, factors that differentiate underserved populations may be geographically distinct, meaning that findings from recent U.S.-based studies may generalize poorly to low- and middle- income countries located, e.g., in Africa or Latin America. Underserved populations in urban settings in these countries typically live in more densely populated areas, both by neighborhood and household assessments; rely mainly or exclusively on crowded public transportation to get around; tend to be employed in public-facing occupations; and have limited access to private health insurance.

Our goal is to contribute to the discussion on COVID-19 disparities by assessing the role of socioeconomic factors in predicting patient outcomes in Brazil, a low- and middle- income country (LMIC). At the time of this report, Brazil presented the second highest number of

total confirmed cases and deaths worldwide [2]. We use a highly representative dataset of COVID-19 patients from Brazil to derive machine learning models that predict in-hospital death and ventilator usage. To the best of our knowledge, this is the first study to evaluate the effect of non-clinical factors, including patients' self-reported race and education level, access to private hospitals, and geographic location of the hospital, on COVID-19 mortality and resource use. Moreover, this is one of the largest datasets used to date, with over 159,000 hospitalized COVID-19 patients, including 54,000 deceased.

To develop the predictive models, we leverage both interpretable machine learning (ML) methods and others which form ensembles of a large number of decision trees and, thus, are not easy to interpret. We find that the simpler interpretable models, coupled with optimized feature selection, perform just as well as the complex non-interpretable models. This contributes to the discussion on using interpretable ML models for high-stake decision-making [21, 22].

## Data

The first confirmed COVID-19 case in Brazil was reported on February 26, 2020 in the state of São Paulo [23, 24]. Starting in March 2020, control measures were implemented in the country in a decentralized manner, with each state being responsible for the adoption and enforcement of its own set of social distancing measures. The states of São Paulo and Rio de Janeiro were the first to shut down non-essential services, including shopping and fitness centers, and to cancel all public events [25]. At the time of this report, just over six months after confirmation of the first case, the total number of cases in Brazil surpassed the 4 million mark, with over 125,500 deaths [1], albeit with an estimated reporting rate of only 9.2% [26].

In 2009, the Brazilian Ministry of Health established a nationwide surveillance program for acute respiratory distress syndrome (ARDS) following the H1N1 Influenza outbreak. The program maintains a publicly available database repository [27] in which all health care institutions must report confirmed ARDS cases. For reporting purposes, Influenza patients are classified as those who present fever or a fever sensation accompanied by one or more of the following symptoms: cough, sore throat, dripping nose, difficulty breathing, and nose running down throat. If the condition of a flu patient develops into one or more of the symptoms below, they are classified as ARDS: dyspnea/respiratory distress, persistent chest pressure, oxygen saturation less than 95% in ambient air, bluish color of the lips or face.

In 2020, the ARDS program was extended to include COVID-19 surveillance. Data used in this study was extracted from the ARDS surveillance database repository (accessed on July 2, 2020), and included information on demographic characteristics, symptoms and comorbidities, resource usage, x-ray thorax results, and COVID-19 outcome (recovered, deceased, ongoing). Because our goal is to generate predictive models for mortality and ventilation need, we filtered the dataset and retained only cases pertaining to hospitalized patients who tested positive for COVID-19 and had either recovered or were deceased between March 1 and June 30, 2020. We removed outliers in the dataset which are easily identified, for example, repeated rows, empty entries, and the pregnancy of male patients. After this cleaning process, the number of patients left was 113,314 including 50,387 deceased. A description of the patient features in the dataset with corresponding counts is provided in Table 1. Note that the sum of the categories of a variable may not total 100%, e.g., in the Race variable. This means that the rest of the observations have unknown values for this variable. In addition to Table 1, Fig 1 shows the fraction of deceased patients across different characteristics and age groups, e.g., in the upper-right box, 0.7 is the ratio of deceased patients who are 65–100 years old and have ARDS over the total number of 65–100 years old patients with ARDS (deceased or not).

**Table 1. Patient characteristics in the dataset reported as: Count (percentage).**

| | | | | |
|---|---|---|---|---|
| Demographics | Gender | Female | 49184 | (43.4%) |
| | | Other | 32 | (0.0%) |
| | | Male | 63998 | (56.5%) |
| | Race | White | 32704 | (28.9%) |
| | | Yellow | 1115 | (1.0%) |
| | | Indigenous | 366 | (0.3%) |
| | | Brown/Black | 40993 | (36.2%) |
| | Schooling | No Education | 2799 | (2.5%) |
| | | Elem 1-5 | 9374 | (8.3%) |
| | | Elem 6-9 | 6727 | (5.9%) |
| | | Medium 1-3 | 12629 | (11.2%) |
| | | Superior | 6572 | (5.8%) |
| | Region | Midwest | 5931 | (5.2%) |
| | | North | 13948 | (12.3%) |
| | | Northeast | 23918 | (21.1%) |
| | | South | 5746 | (5.1%) |
| | | Southeast | 63671 | (56.2%) |
| | Age | 0-30 | 7474 | (6.6%) |
| | | 30-50 | 29032 | (25.6%) |
| | | 50-65 | 31280 | (27.6%) |
| | | 65-100 | 45233 | (40.0%) |
| Symptoms | | Fever | 80530 | (71.1%) |
| | | Cough | 84803 | (74.9%) |
| | | Throat | 22902 | (20.2%) |
| | | Dyspnea | 79933 | (70.6%) |
| | | Respiratory Discomfort | 64854 | (57.3%) |
| | | SpO2 less 95% | 62908 | (55.6%) |
| | | Diarrhea | 15493 | (13.7%) |
| | | Vomiting | 8753 | (7.7%) |
| | | Other Symptoms | 37791 | (33.4%) |
| Prior Medical Conditions | | Postpartum | 387 | (0.3%) |
| | | Cardiovascular Disease | 37392 | (33.0%) |
| | | Hematologic Disease | 1052 | (0.9%) |
| | | Down Syndrome | 298 | (0.3%) |
| | | Liver Chronic Disease | 1068 | (0.9%) |
| | | Asthma | 3046 | (2.7%) |
| | | Diabetes | 29120 | (25.7%) |
| | | Neurological Disease | 4516 | (4.0%) |
| | | Another Chronic Pneumopathy | 4281 | (3.8%) |
| | | Immunosuppression | 3455 | (3.1%) |
| | | Renal Chronic Disease | 4945 | (4.4%) |
| | | Obesity | 4186 | (3.7%) |
| | | Other Risks | 30105 | (26.6%) |
| COVID-19 related | Resources | Antiviral Use | 33785 | (29.8%) |
| | | ICU | 35675 | (31.5%) |
| | | Ventilator Invasive | 22571 | (19.9%) |

(*Continued*)

**Table 1.** (Continued)

| | Xray Thorax Result | Normal | 2892 | (2.6%) |
|---|---|---|---|---|
| | | Interstitial infiltrate | 20600 | (18.2%) |
| | | Consolidation | 3077 | (2.7%) |
| | | Mixed | 3678 | (3.2%) |
| | | Other | 18956 | (16.7%) |
| | Outcome | Recovered | 62827 | (55.5%) |
| | | Deceased | 50387 | (44.5%) |
| | Hospital | Public | 22745 | (20.1%) |
| | | Private | 28041 | (24.8%) |
| | Other | Acute Respiratory Distress Syndrome | 28496 | (25.2%) |
| | | Contracted At Hospital | 2687 | (2.4%) |

## Methods

The study analyzed publicly available data that have been fully de-identified, so additional ethical approval was not required. The primary objective in learning a classifier is to maximize prediction accuracy (or minimize a loss function). In light of the discussion on favoring interpretable models, we will examine our models from two aspects: prediction performance and interpretability.

## Classifiers

We are interested in defining two prediction tasks, mortality and the need for mechanical ventilation. For each task, we build five classifiers using Logistic Regression (LR), sparse versions of LR and Support Vector Machines (SVM), Random Forests [28], and Gradient Boosted Trees (XGBoost [29]). We choose to construct the SVM and LR classifiers given their ability to provide quantifiable associations with specific variables driving the predictions, which is critical in our setting. Conversely, we use state-of-the-art algorithms: Random Forests and XGBoost, to compare their performance with LR and SVM. A brief discussion of these methods is provided in the S1 File.

Evidence has shown that *sparse* classifiers, i.e., the ones which use a parsimonious set features, offer higher interpretability and they perform better out of sample [30]. To that end, we develop a fully automated pre-processing procedure to select a smaller subset of variables to be used in the classification task. The steps we employ are as follows.

## Pre-processing and feature selection

First, we (i) *remove unknown or missing entries*: After performing one-hot encoding for categorical features, we discard all the new variables corresponding to *unknown* or *missing* entries, given that these do not add any new information to our predictive task and harm interpretability. Then, we (ii) *remove correlated variables* to avoid collinearity. In particular, we calculate pairwise correlations among variables, and remove one variable from each highly correlated pair (those with an absolute correlation coefficient greater than 0.8). Next, we (iii) *remove low influence variables*: we separate observations in two classes, the positive (e.g., deceased, or ventilated) and the negative class. Then, for each feature we test whether the two cohorts have the same mean by performing a two-sided *t*-test. To keep the variables with the higher impact, we retain the ones for which we have a 95% confidence that the mean for the two samples is different. Finally, we perform (iv) *Cross-Validated Recursive Feature Elimination* [31]: this

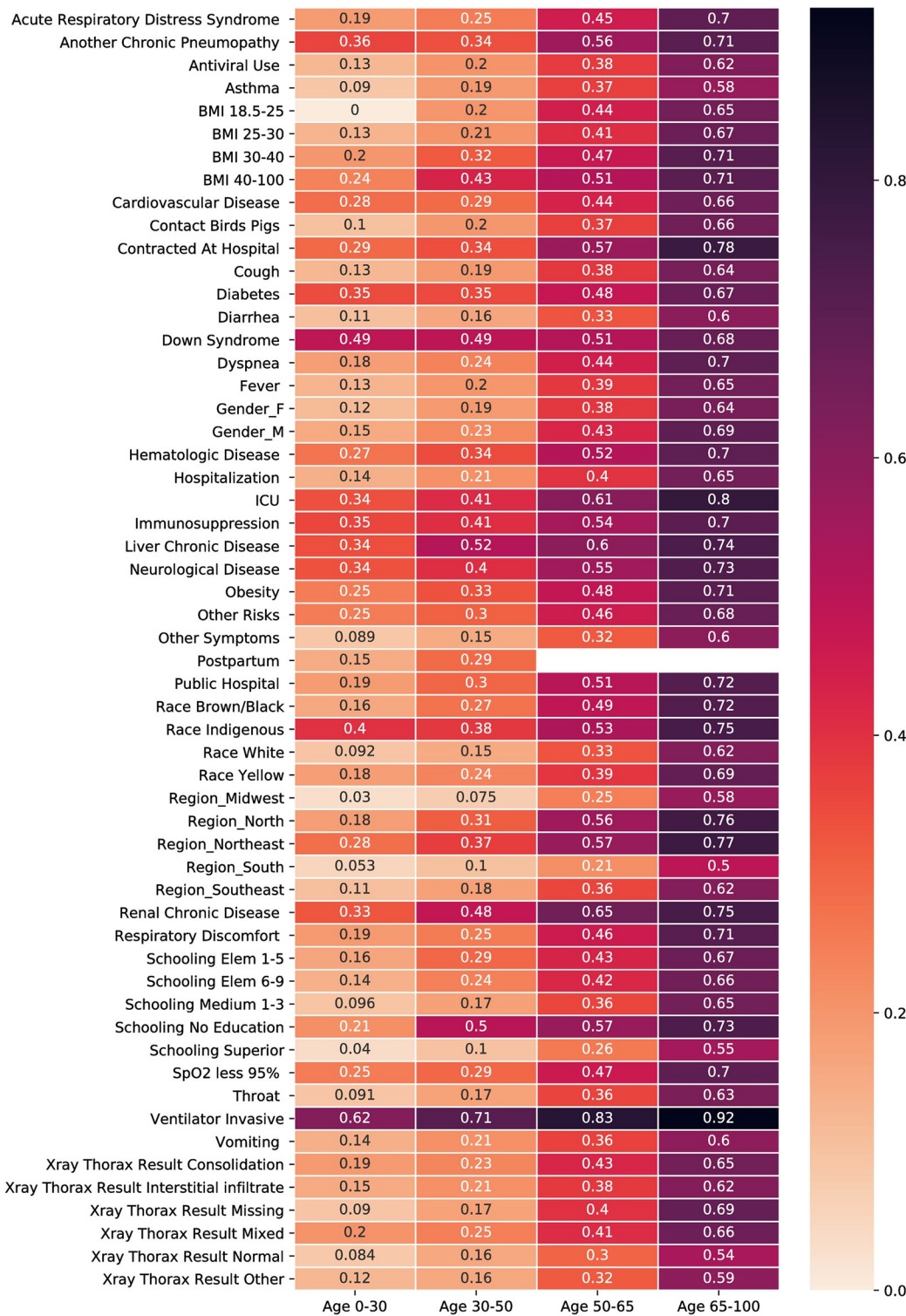

**Fig 1. Fraction of deceased patients given a certain feature and age group.**

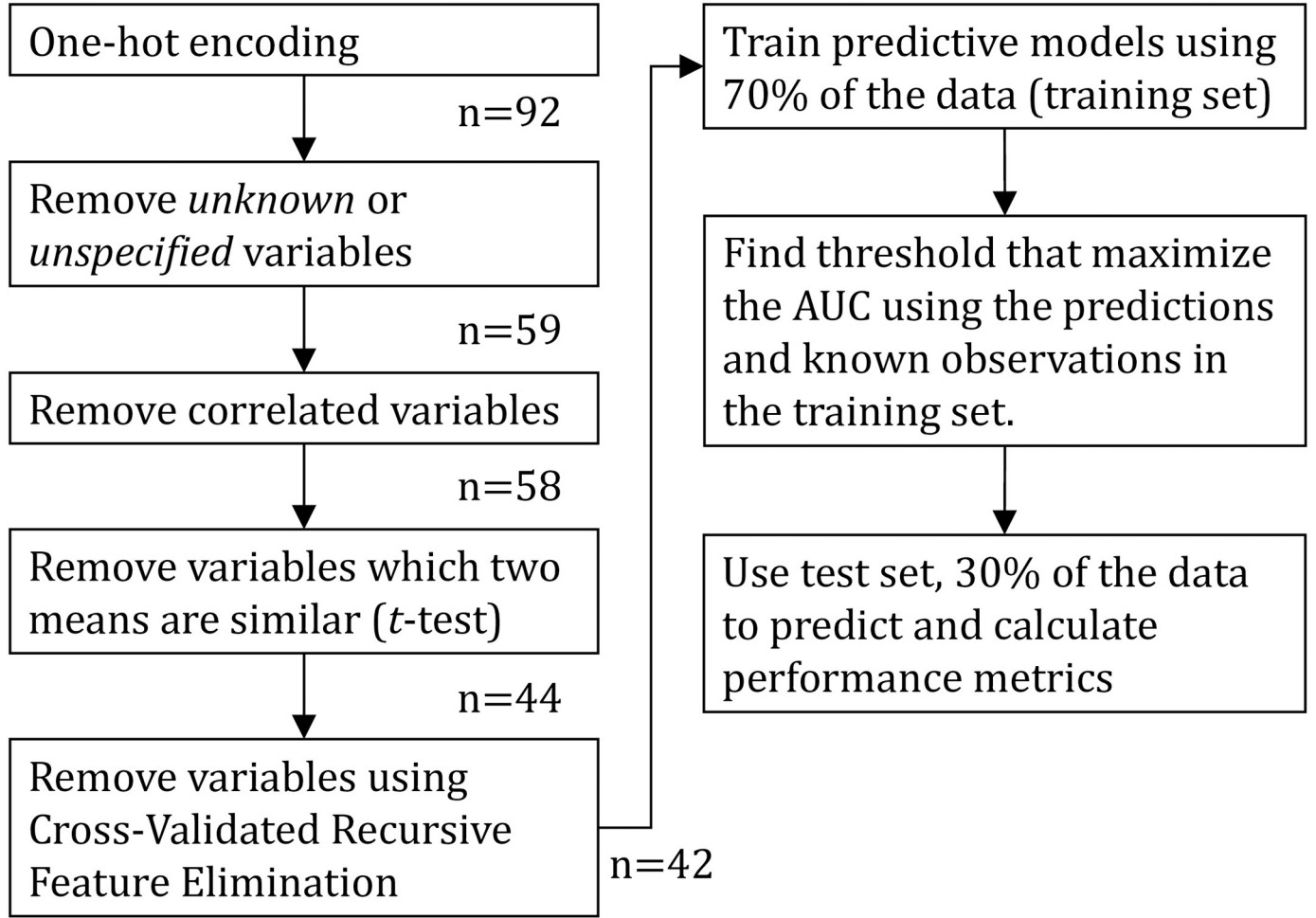

**Fig 2. Flow diagram describing the general procedure employed in this paper.** *n* is the number of variables available after each step of the pipeline for the mortality model.

procedure begins by learning a classifier (we use LR) using all features and computing an importance score. For LR, the importance score is the (absolute) magnitude of the linear coefficient $\beta_i$ of feature $i$. After this step, the least important feature (the one with the smallest $|\beta_i|$) is deleted, and this process is repeated iteratively until a single variable is left. At each iteration, we report the performance of the model by using a ten-fold cross-validation, and we pick the set of features that maximize this value. A summary of this feature selection procedure is presented in Fig 2. Note that normalization is not needed given that we are using only binary variables.

### Performance evaluation and validation

For all models, we split patients into a training (70%) and test set (30%). We train the models on the training test, and report performance metrics on the test set (out-of-sample). Fig 2 sketches the full approach employed in this paper. To evaluate the performance of the trained classifiers we use two metrics: the false alarm (or false positive) rate and the detection rate. The false alarm rate is the fraction of the patients predicted to be in the positive class while they truly were not, among all negative class patients. The term *specificity* is often used and it equals

1 minus the false alarm rate. In turn, the detection rate measures the number of patients predicted to be in the positive class while they truly were, divided by all positive class patients. In the medical literature, the detection rate is often referred to as *sensitivity* or *recall*. A single metric that encapsulates these errors is the Area Under the Curve (AUC) of the Receiver Operating Characteristic (ROC). The ROC plots the detection rate over the false positive rate. A blind random selection (assigning patients to classes randomly) has AUC of 0.5 while a perfect classifier an AUC of 1. In addition to the AUC, we report the accuracy of the classifier which calculates the ratio between the number of correct classifications over the total number of predictions. Moreover, we report the weighted F1-score to summarize the *precision* and *recall* for both the positive class and the negative class. The weighted F1-score (F1w) computes the weighted average (using the number of samples per class) of the harmonic mean of *precision* and *recall* per class. This metric is of interest to this work because it is as important to accurately predict who is likely to, or not to, have a specific outcome. For example, one can lessen physical distancing restrictions based on those who are predicted to have lower risk.

## Results

We train two classifiers using 70% of the observations to predict (1) mortality and (2) need for a mechanical ventilator for a COVID-19 patient based on demographics, comorbidities, symptoms, and some clinical information (e.g., x-ray findings). For each model, we compare the performance of five different predictors, which include interpretable and non-interpretable state-of-the-art classifiers. Our results suggest that LR and SVM achieve comparable performance to the non-interpretable methods, as can be seen in Tables 2 and 3, and provide insights about how different features affect the outcome. Observe that the more complicated methods, RF and XGBoost, do not provide any improvement in performance compared with LR for both tasks.

As mentioned earlier, interpretability is desired in this application to identify the main variables used to classify an individual as high (or low) risk. This information can be obtained through the coefficients of the LR model, the odds ratio (OR) and the corresponding confidence intervals (CI) obtained for each variable. Specifically, the Odds Ratio (OR) indicates how the odds of observing the outcome are scaled when the variable takes the value 1 (vs. 0), while controlling for all other variables in the model. Once we identify the features to be used from our feature selection procedure, we use $\ell_2$-regularized LR to compute the coefficients, ORs, and the corresponding confidence intervals.

**Table 2. Mortality results.**

|          | SVM-l1 | LR-l1 | LR-l2 | RF    | XGBoost |
|----------|--------|-------|-------|-------|---------|
| Accuracy | 0.720  | 0.719 | 0.718 | 0.713 | 0.719   |
| F1w      | 0.721  | 0.720 | 0.719 | 0.714 | 0.719   |
| AUC      | 0.790  | 0.790 | 0.790 | 0.786 | 0.792   |

**Table 3. Ventilator results.**

|          | SVM-l1 | LR-l1 | LR-l2 | RF    | XGBoost |
|----------|--------|-------|-------|-------|---------|
| Accuracy | 0.764  | 0.766 | 0.766 | 0.763 | 0.761   |
| F1w      | 0.746  | 0.746 | 0.745 | 0.747 | 0.745   |
| AUC      | 0.694  | 0.694 | 0.694 | 0.695 | 0.695   |

Some of the main features that predict mortality and the need for a mechanical ventilator are related with socioeconomic characteristics rather than with prior medical conditions or symptoms (see Table 4 and Figs 3 and 4), which can motivate further investigation in this direction. We observe that for predicting mortality, geographic location of the hospital (Northeast OR = 2.2, North OR = 2.0, Midwest OR = 0.8, South OR = 0.6), education level (No education OR = 1.1, Elementary 1-5 OR = 1.0, Medium 1-3 OR = 0.9, Superior OR = 0.6), hospital type (Public OR = 1.24, Private OR = 0.65), and race (Indigenous OR = 1.2, Yellow OR = 1.2, White OR = 0.9) are key variables for classifying the outcome of a patient. Furthermore, to predict the need for mechanical ventilation, geographic location (Northeast OR = 0.53, Midwest OR = 0.45, South OR = 0.33, Southeast OR = 0.33) and education level (Medium 1-3 OR = 0.77, Superior OR = 0.71) are relevant variables. From a clinical perspective, the results of the coefficients are consistent with recent studies highlighting the importance of variables such as age, chronic renal insufficiency, hypoxia, diabetes, and obesity. Figs 3 and 4 depict the ORs with their confidence intervals for the mortality and ventilator models respectively.

In addition to these two models, we train more *advanced* models for predicting the events of interest. These *advanced* models are provided with more information about the evolution of the disease. For mortality, we include information on whether a patient is in an ICU and on a ventilator. When these data is provided, the accuracy and AUC of the model increases by 6.8% and 8.0%, respectively, compared to the ones presented in Table 2 and Fig 3. Conversely, for the *advanced* ventilation model, we include the variable ICU which improves the accuracy and AUC of the model by 8.7% and 24.6% respectively. The specific results of these models are provided in the S1 File of this manuscript.

## Discussion

We generated moderately to significantly accurate predictive models of mortality and ventilator use for COVID-19 patients that are sparse and interpretable based only on demographics, symptoms, comorbidities, and socieconomic variables. Our results confirm previously described clinical presentations and outcomes of COVID-19-related hospital admissions, but also suggest that additional non-clinical features, in particular sociodemographic information, are important explanatory variables. The following comorbidities were found to be highly predictive of mortality: renal (OR = 2.0) and liver chronic disease (OR = 1.7), immunosuppression (OR = 1.7), obesity (OR = 1.7), chronic pneumopathy (OR = 1.4), neurological (OR = 1.6), hematologic (OR = 1.2) and cardiovascular (OR = 1.1) disease, diabetes (OR = 1.4), and immunosuppression (OR = 1.3). Respiratory symptoms, ranging from respiratory discomfort (OR = 1.4) and dyspnea (OR = 1.32) to oxygen saturation less than 95% (OR = 1.7), were also significantly associated with mortality risk among COVID-19 patients. Of note, cardiovascular disease includes hypertension, history of myocardial infarction, stroke, congestive heart failure, and other forms of heart disease. Its low effect on predicting mortality is consistent with the observations in [32].

Unlike previous studies, we assessed the relationship between socioeconomic factors and mortality and resource utilization in a low- and middle- income country (LMIC), and found low patient-reported level of education to be significantly associated with mortality (See Table 4). We observe that OR for mortality is inversely proportional to self-reported education level, which is suggestive of disparity on health outcomes for different population subgroups. A 2017 census revealed that 7% of the population aged 15 years or older in Brazil was illiterate [33]; this corresponds to approximately 11.5 million inhabitants. In addition to education, we found that geographic location of the hospital in which a COVID-19 patient was admitted was also a strong predictor of outcome. Based on postal code, we mapped hospital location to one

**Table 4. Mortality coefficients for $\ell_2$-LR.**

|  | β | CI (2.5) | CI (97.5) | OR | CI (2.5) | CI (97.5) |
|---|---|---|---|---|---|---|
| Age 0-30 | -1.988 | -2.413 | -1.562 | 0.137 | 0.090 | 0.210 |
| Age 30-50 | -1.426 | -1.843 | -1.008 | 0.240 | 0.158 | 0.365 |
| Region_Northeast | 0.782 | 0.362 | 1.201 | 2.185 | 1.436 | 3.325 |
| Region_North | 0.723 | 0.302 | 1.144 | 2.061 | 1.352 | 3.140 |
| Age 50-65 | -0.712 | -1.129 | -0.295 | 0.491 | 0.323 | 0.745 |
| Renal Chronic Disease | 0.694 | 0.611 | 0.776 | 2.001 | 1.842 | 2.173 |
| Contracted At Hospital | 0.591 | 0.477 | 0.704 | 1.805 | 1.612 | 2.023 |
| Liver Chronic Disease | 0.540 | 0.366 | 0.714 | 1.716 | 1.442 | 2.041 |
| Immunosuppression | 0.511 | 0.413 | 0.609 | 1.667 | 1.512 | 1.838 |
| Obesity | 0.510 | 0.421 | 0.598 | 1.665 | 1.524 | 1.819 |
| SpO2 less 95% | 0.504 | 0.466 | 0.543 | 1.656 | 1.594 | 1.721 |
| Neurological Disease | 0.496 | 0.410 | 0.582 | 1.642 | 1.507 | 1.790 |
| Cough | -0.485 | -0.527 | -0.443 | 0.616 | 0.590 | 0.642 |
| Region_South | -0.481 | -0.909 | -0.054 | 0.618 | 0.403 | 0.947 |
| Schooling Superior | -0.464 | -0.551 | -0.377 | 0.629 | 0.577 | 0.686 |
| Hospital Private | -0.428 | -0.470 | -0.385 | 0.652 | 0.625 | 0.681 |
| Other Symptoms | -0.419 | -0.456 | -0.381 | 0.658 | 0.634 | 0.683 |
| Down Syndrome | 0.388 | 0.072 | 0.705 | 1.475 | 1.075 | 2.023 |
| Another Chronic Pneumopathy | 0.358 | 0.270 | 0.445 | 1.430 | 1.310 | 1.561 |
| Respiratory Discomfort | 0.301 | 0.262 | 0.339 | 1.351 | 1.300 | 1.404 |
| Dyspnea | 0.279 | 0.238 | 0.321 | 1.322 | 1.268 | 1.379 |
| Other Risks | 0.262 | 0.223 | 0.300 | 1.299 | 1.250 | 1.350 |
| Diarrhea | -0.257 | -0.310 | -0.205 | 0.773 | 0.733 | 0.815 |
| Fever | -0.249 | -0.289 | -0.208 | 0.780 | 0.749 | 0.812 |
| Age 65-100 | 0.244 | -0.173 | 0.660 | 1.276 | 0.841 | 1.936 |
| Asthma | -0.229 | -0.339 | -0.120 | 0.795 | 0.713 | 0.887 |
| Gender_F | -0.216 | -0.251 | -0.181 | 0.806 | 0.778 | 0.834 |
| Hospital Public | 0.214 | 0.171 | 0.257 | 1.239 | 1.187 | 1.293 |
| Diabetes | 0.198 | 0.159 | 0.238 | 1.219 | 1.172 | 1.269 |
| Throat | -0.191 | -0.236 | -0.145 | 0.827 | 0.790 | 0.865 |
| Region_Midwest | -0.178 | -0.608 | 0.252 | 0.837 | 0.545 | 1.286 |
| Hematologic Disease | 0.173 | -0.002 | 0.348 | 1.188 | 0.998 | 1.416 |
| Race Indigenous | 0.167 | -0.139 | 0.472 | 1.181 | 0.871 | 1.603 |
| Schooling Medium 1-3 | -0.154 | -0.213 | -0.095 | 0.858 | 0.808 | 0.910 |
| Race Yellow | 0.145 | -0.022 | 0.312 | 1.156 | 0.978 | 1.366 |
| Cardiovascular Disease | 0.113 | 0.075 | 0.151 | 1.120 | 1.078 | 1.163 |
| Xray Thorax Result Consolidation | 0.105 | 0.004 | 0.206 | 1.111 | 1.004 | 1.229 |
| Acute Respiratory Distress Syndrome | 0.075 | 0.035 | 0.114 | 1.078 | 1.036 | 1.121 |
| Postpartum | 0.067 | -0.258 | 0.391 | 1.069 | 0.773 | 1.479 |
| Schooling No Education | 0.064 | -0.049 | 0.178 | 1.066 | 0.952 | 1.194 |
| Race White | -0.062 | -0.104 | -0.020 | 0.940 | 0.902 | 0.980 |
| Vomiting | -0.055 | -0.123 | 0.012 | 0.946 | 0.885 | 1.012 |
| Region_Southeast | 0.029 | -0.390 | 0.449 | 1.030 | 0.677 | 1.566 |
| Schooling Elem 1-5 | -0.017 | -0.078 | 0.044 | 0.984 | 0.925 | 1.045 |

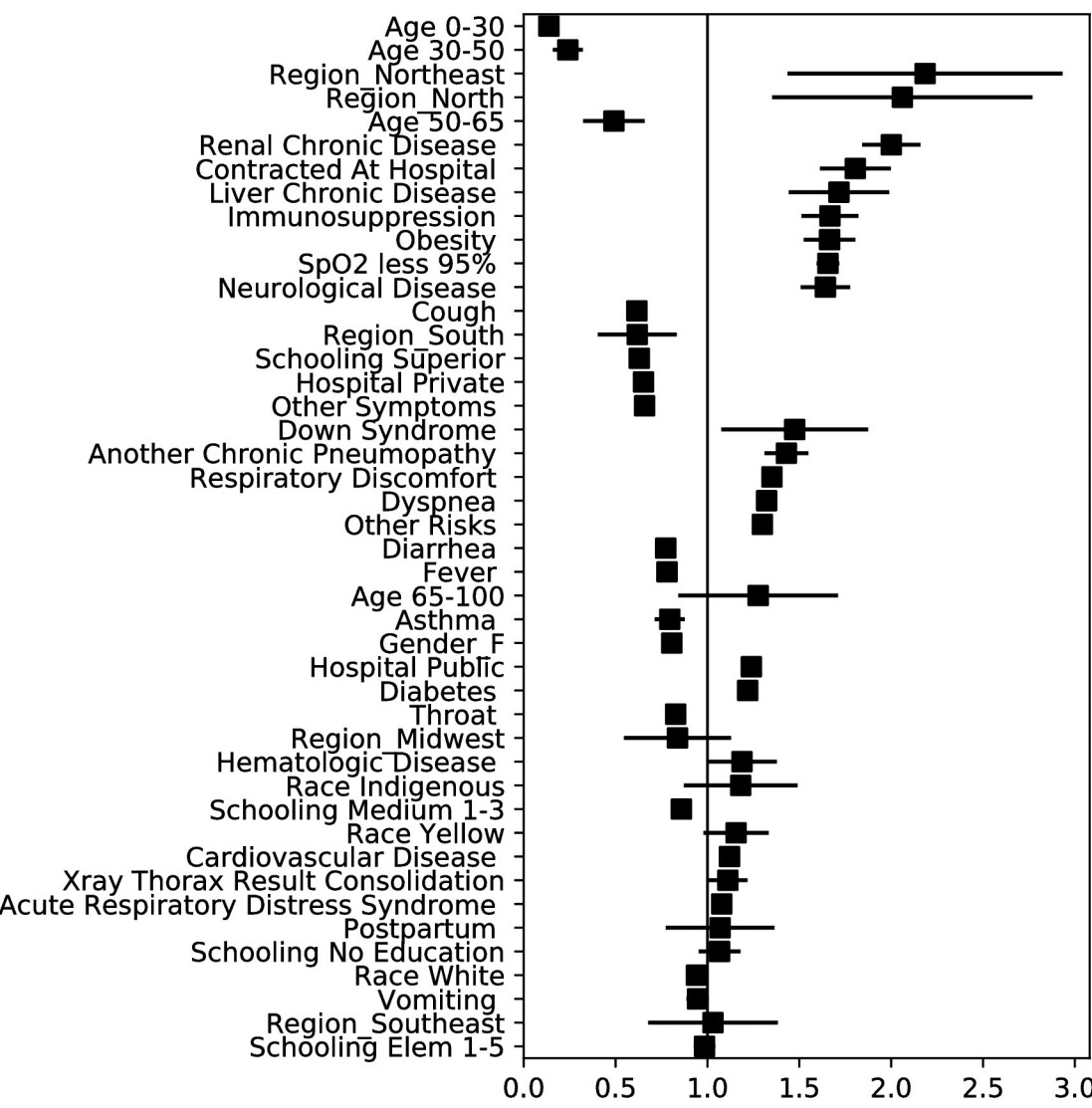

**Fig 3. Odds ratios and confidence intervals for the $\ell_2-$LR mortality model.**

of five geopolitical regions of Brazil: North, Northeast, Midwest, Southeast, and South. Although these regions are officially recognized, this division has no political effect other that guiding the development of federal public policies. Currently, patterns of economic activity and population settlement vary widely among the regions, as do development indices. The average Human Development Index (HDI) in North and Northeastern regions is significantly lower than the national average (0.66 in both regions vs. 0.76 nationwide), as are the literacy rates. In this context, it is revealing that the odds of mortality to COVID-19 were significantly higher for patients hospitalized in the North and Northeast regions.

The Unified Health System (Sistema Único de Saúde—SUS), Brazil's publicly funded health care system, was created by a constitutional act in 1989. It represents the only source of medical care for approximately 75% of the population [34], 80% of which are of self-reported black race [35]. Although Brazil has a mixed delivery system of public and private hospitals, only 24.2% of the population has private insurance [36]. As in many LMICs, SUS is underfunded and overstretched, and resource availability in public health care institutions is limited in

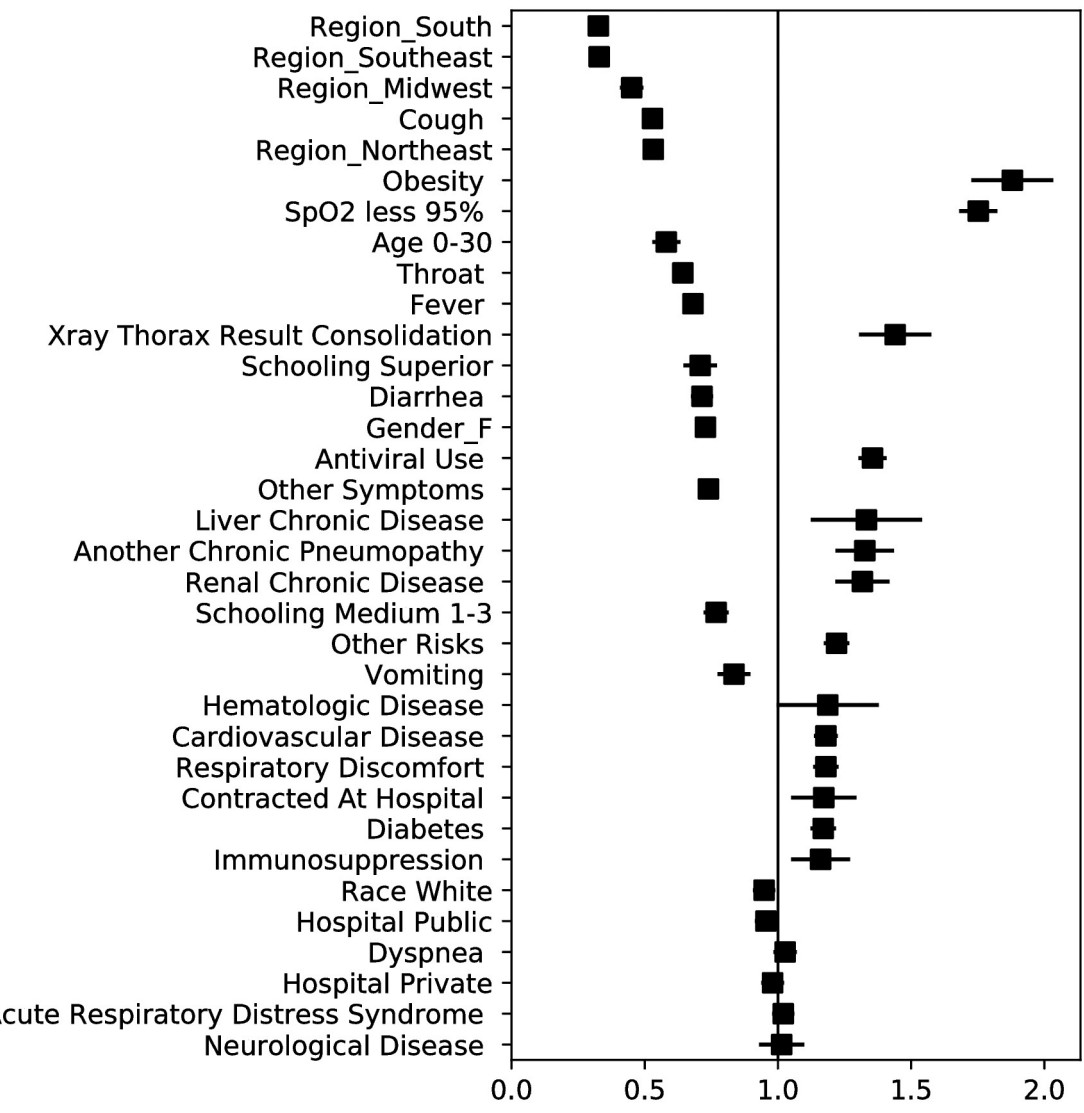

**Fig 4. Odds ratios and confidence intervals for the $\ell_2$−LR ventilator model.**

comparison with their private counterparts [37]. In contrast, large public hospitals serve as the entry point into the health care system for many severe and/or urgent patients, including those who have access to private insurance and are frequently transferred to private hospitals following initial assessment. Interestingly, our results indicate that COVID-19 patients hospitalized in public hospitals have higher risk of mortality, irrespective of the geographic location of the hospital (as we are controlling for this variable).

Taken together, our results highlight the predictive power of socioeconomic information in assessing COVID-19 mortality. From a practical perspective, our findings suggest that decisions on medical resource allocation throughout the COVID-19 pandemic could be guided by local patterns of patient demographics within a LMIC. Moreover, our study suggests that the definition of vulnerable subgroups, for the purposes of targeted policy design, encompasses not only individual patient features (such as race and education level), but also an understanding of the structure of the health care system by which these patients are served.

## Study limitations

First, we do not claim our results to provide a complete causal-effect analysis, as this task requires a more sophisticated analysis. However, we do think that given all the controls in our models, these results shed light and motivate further investigations of social disparities in health care access in LMICs. Second, from a clinical point of view, it is relevant to highlight that the dataset lacks important information (such as lab results) to provide a clinical assessment of COVID-19. Such information is hard to obtain at the scale we consider. Rather, the focus of this work is to open the discussion about socioeconomic disparities in health access, as well as to help inform decisions on how to best allocate limited medical resources and design targeted policies for vulnerable subgroups which might not have access to clinical and lab assessments. Third, we note that the dataset might be biased towards assessing the risk of high-risk patients given that we are observing only COVID-19 cases which have been hospitalized. for this study dataset does not include specific dates at which hospitals discharge patients, which is of high importance to assess the utilization of medical equipment. to prioritize the use of resources, we understand that medical risk is not the only factor in making such decisions. Nevertheless, in order to quantify medical risk one can leverage the models presented in this work.

## Conclusions

Classifying the medical risk of COVID-19 patients is relevant for low- and medium- income countries in order to assign limited medical resources more effectively, as well as to help design targeted physical-distancing and work accommodation policies that will assist in reducing economic loss during the current pandemic. In the future, this model could help prioritize vaccine distribution to the more risk-vulnerable and to those who need to interact with them.

To facilitate further work, and for the sake of reproducibility, our models and results are available on a public repository [38].

## Supporting information

**S1 File.**
(ZIP)

## Author Contributions

**Conceptualization:** Salomón Wollenstein-Betech, Julia L. Fleck, Christos G. Cassandras, Ioannis Ch. Paschalidis.

**Data curation:** Salomón Wollenstein-Betech, Amanda A. B. Silva.

**Formal analysis:** Salomón Wollenstein-Betech.

**Funding acquisition:** Julia L. Fleck, Christos G. Cassandras, Ioannis Ch. Paschalidis.

**Investigation:** Amanda A. B. Silva, Julia L. Fleck, Ioannis Ch. Paschalidis.

**Methodology:** Salomón Wollenstein-Betech, Ioannis Ch. Paschalidis.

**Project administration:** Ioannis Ch. Paschalidis.

**Resources:** Julia L. Fleck, Christos G. Cassandras, Ioannis Ch. Paschalidis.

**Software:** Salomón Wollenstein-Betech.

**Supervision:** Christos G. Cassandras, Ioannis Ch. Paschalidis.

**Writing – original draft:** Salomón Wollenstein-Betech, Julia L. Fleck, Ioannis Ch. Paschalidis.

**Writing – review & editing:** Salomón Wollenstein-Betech, Amanda A. B. Silva, Julia L. Fleck, Christos G. Cassandras, Ioannis Ch. Paschalidis.

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
