## [Decision Letter · Decision Letter 0]

3 Sep 2020

PONE-D-20-22853

Physiological and Socioeconomic Characteristics Predict COVID-19 Mortality and Resource Utilization in Brazil

PLOS ONE

Dear Dr. Paschalidis,

Thank you for submitting your manuscript to PLOS ONE. After careful consideration, we feel that it has merit but does not fully meet PLOS ONE’s publication criteria as it currently stands. Therefore, we invite you to submit a revised version of the manuscript that addresses the points raised during the review process.

ACADEMIC EDITOR: I have received the comments of the reviewers on your manuscript. The specific comments of the reviewers are included below. Please provide point by point response in your revised manuscript.

We look forward to receiving your revised manuscript.

Kind regards,

Muhammad Adrish

Academic Editor

PLOS ONE

Journal Requirements:

2. Thank you for including your ethics statement:  "The study analyzed publicly available data that have been fully de-identified, so it is not considered human subject research. " Please revise this to state ""The study analyzed publicly available data that have been fully de-identified, so additional ethical approval was not required.

3. Please include the date(s) on which you accessed the databases or records to obtain the data used in your study.

4. For studies involving humans categorized by race/ethnicity, age, disease/disabilities, religion, sex/gender, sexual orientation, or other socially constructed groupings, authors should:

1) Explicitly describe their methods of categorizing human populations,

2) Define categories in as much detail as the study protocol allows,

3) Justify their choices of definitions and categories,

4) Explain whether (and if so, how) they controlled for confounding variables such as socioeconomic status, nutrition, environmental exposures, or similar factors in their analysis, and

5) Update outmoded terms and potentially stigmatizing labels to more current, acceptable terminology.

Examples: “Caucasian” should be changed to “white” or “of [Western] European descent” (as appropriate); “XXX victims” should be changed to “patients with XXX.

5. Please ensure that you refer to Figure 4 in your text as, if accepted, production will need this reference to link the reader to the figure.

6. Please provide additional details regarding participant consent. In the Methods section, please ensure that you have specified (1) whether consent was informed and (2) what type you obtained (for instance, written or verbal). If your study included minors, state whether you obtained consent from parents or guardians. If the need for consent was waived by the ethics committee, please include this information.

Reviewers' comments:

Reviewer's Responses to Questions

**Comments to the Author**

1. Is the manuscript technically sound, and do the data support the conclusions?

Reviewer #1: Yes

Reviewer #2: Yes

2. Has the statistical analysis been performed appropriately and rigorously? 

Reviewer #1: Yes

Reviewer #2: Yes

3. Have the authors made all data underlying the findings in their manuscript fully available?

Reviewer #1: Yes

Reviewer #2: Yes

4. Is the manuscript presented in an intelligible fashion and written in standard English?

Reviewer #1: Yes

Reviewer #2: Yes

5. Review Comments to the Author

Reviewer #1: I read with great interest this manuscript

I find it well wrote and wiht high research quality from a country that need also scientific attention

Only some suggestions:

1. Introduction: update data of COVID cases in Brazil at revision's day

2. Methods and results: very well wrote

3.Discussion: discuss better the role of cardiovascolar risk factor on outcome (see and cite https://doi.org/10.1016/j.numecd.2020.07.031) and future perspective from your data (see and cite doi:10.3390/ijerph17082690)

Conclusion: They are coherent with the manuscript

I appreciate your manuscript and find tables and statistical analisys very well done

Reviewer #2: The manuscript entitled “Physiological and Socioeconomic Characteristics Predict COVID-19 Mortality and Resource Utilization in Brazil” by Wollenstein-Betech et al, performed statistical analysis on publicly available data of 113,214 patients, including 50,387 deceased. Authors built 5 different classifiers, LR, sparse version of LR, SVM, RF and XGBoost to predict mortality and the need for mechanical ventilators for a COVID-19 patient using demographics, comorbidities, symptoms, and some clinical information. They construct SVM and LR classifiers and compare their performance with RF and XGBoost and found that LR and SVM achieve comparable performance. The manuscript is well designed and provides lots of useful information. However, the result section needs improvement.

The study shows that the death from COVID-19 was strongly associated with demographics, socioeconomic factors, and comorbidities. It is well known that mortality in the case of COVID-19 condition is strongly associated with comorbidities. The new information provided by this study is its association with demographics and the socioeconomic factors. However, authors have not put the details in the result section or discussion. The result section can be strengthen. Lots of data have been provided in the tables/ figures which are not mentioned in the result section. For better understanding of the manuscript by the readers, authors need to include these data in the result section.

Although mentioned in the tables and figures, authors need to discuss more about the educational level and access to private hospital in the discussion section. Ultimate aim of the manuscript is to predict mortality and the need for mechanical ventilators for a COVID-19 patient using demographics, comorbidities, symptoms, and some clinical information. Authors should provide their suggestions about “how these parameters can be used by the healthcare professionals”.

One of the major cause for the mortality could be longer access to the healthcare and late reporting of the case. Can authors provide any data (if possible) and mention it in the result/ discussion section.

Table 1: Why in the demographics section schooling and region are not equating to 100%

Line 214: check the sentence

6. PLOS authors have the option to publish the peer review history of their article (what does this mean?). If published, this will include your full peer review and any attached files.

Reviewer #1: **Yes: **Francesco Di Gennaro

Reviewer #2: No

---

## [Author Response · Author response to Decision Letter 0]

18 Sep 2020

Please see attached Response to Reviews document.

---

## [Decision Letter · Decision Letter 1]

25 Sep 2020

Physiological and Socioeconomic Characteristics Predict COVID-19 Mortality and Resource Utilization in Brazil

PONE-D-20-22853R1

Dear Dr. Paschalidis,

We’re pleased to inform you that your manuscript has been judged scientifically suitable for publication and will be formally accepted for publication once it meets all outstanding technical requirements.

Kind regards,

Muhammad Adrish

Academic Editor

PLOS ONE

Additional Editor Comments (optional):

Reviewers' comments:

Reviewer's Responses to Questions

**Comments to the Author**

1. If the authors have adequately addressed your comments raised in a previous round of review and you feel that this manuscript is now acceptable for publication, you may indicate that here to bypass the “Comments to the Author” section, enter your conflict of interest statement in the “Confidential to Editor” section, and submit your "Accept" recommendation.

Reviewer #1: All comments have been addressed

Reviewer #2: All comments have been addressed

2. Is the manuscript technically sound, and do the data support the conclusions?

Reviewer #1: Yes

Reviewer #2: Yes

3. Has the statistical analysis been performed appropriately and rigorously? 

Reviewer #1: Yes

Reviewer #2: I Don't Know

4. Have the authors made all data underlying the findings in their manuscript fully available?

Reviewer #1: Yes

Reviewer #2: Yes

5. Is the manuscript presented in an intelligible fashion and written in standard English?

Reviewer #1: Yes

Reviewer #2: Yes

6. Review Comments to the Author

Reviewer #1: Authors improve their manuscript and I find it well done

Tha article, in my opinion, now can be pubblish

Reviewer #2: All the queries raised are satisfactorily answered by the authors. The manuscript may be considered for publication.

7. PLOS authors have the option to publish the peer review history of their article (what does this mean?). If published, this will include your full peer review and any attached files.

Reviewer #1: No

Reviewer #2: No

---

## [Editor Report · Acceptance letter]

7 Oct 2020

PONE-D-20-22853R1 

Physiological and socioeconomic characteristics predict COVID-19 mortality and resource utilization in Brazil 

Dear Dr. Paschalidis:

I'm pleased to inform you that your manuscript has been deemed suitable for publication in PLOS ONE. Congratulations! Your manuscript is now with our production department. 

Kind regards, 

on behalf of

Dr. Muhammad Adrish 

Academic Editor

PLOS ONE